# Multigrid-augmented deep learning preconditioners for the Helmholtz equation

**Yael Azulai**
Department of Computer Science
Ben-Gurion University of the Negev
Beer-Sheva, Israel
yaelaz@post.bgu.ac.il

**Eran Treister**[*]
Department of Computer Science
Ben-Gurion University of the Negev
Beer-Sheva, Israel
erant@cs.bgu.ac.il

## Abstract

We present a data-driven approach to iteratively solve the discrete heterogeneous Helmholtz equation at high wavenumbers. We combine multigrid ingredients with convolutional neural networks (CNNs) to form a preconditioner which is applied within a Krylov solver. Two types of preconditioners are proposed 1) U-Net as a coarse grid solver, and 2) U-Net as a deflation operator with shifted Laplacian V-cycles. The resulting CNN preconditioner can generalize over residuals and a relatively general set of wave slowness models. On top of that, we offer an encoder-solver framework where an "encoder" network generalizes over the medium and sends context vectors to another "solver" network, which generalizes over the right-hand-sides. We show that this option is more efficient than the stand-alone variant. Lastly, we suggest a mini-retraining procedure, to improve the solver after the model is known. We demonstrate the efficiency and generalization abilities of our approach on a variety of 2D problems.

## 1 Introduction

The efficient numerical solution of the Helmholtz equation with a high and spatially-dependent wavenumber is a difficult task. The linear system that results from the discretization of the Helmholtz equation involves a large, complex-valued, sparse, and indefinite matrix. Over the years, various iterative methods were developed for solving the problem, e.g., [18, 7, 8, 25, 5, 27, 6, 16, 21], yet it remains expensive to solve. One of the most common methods to solve it is the Shifted Laplacian (SL) multigrid method [6, 28], which is based on a multigrid preconditioner for a damped version of the equation. That is, a complex shift is added to the matrix to form a preconditioner that is efficiently solved by multigrid components. The preconditioning is applied with a Krylov solver, such as GMRES [23]. However, the SL preconditioner is not scalable due to the high shift that is required to solve the shifted system, and further investigation is needed.

In recent years, Deep Learning (DL) methods have revolutionized many fields, like computer vision and natural language processing [14]. Similarly, solving partial differential equations (PDEs) with a deep neural network has been recently considered in two main forms. One common class of DL-for-PDEs approaches, which are often referred to as physics-informed neural networks (PINNs), includes both PDE solvers [19, 20, 15, 9, 1], and PDE discovery [2]. The general idea is to implement a neural network that implicitly represents the PDE solution. That is, the solution is approximated by the neural network as an analytical function at a given point in space. This function is obtained using stochastic optimization for learning the network's weights as an unsupervised learning task, given a large training set of points. The loss function that is minimized forces the network to meet both the

---

[*]This research was supported by The Israel Science Foundation (grant No. 1589/19).

35th Conference on Neural Information Processing Systems (NeurIPS 2021), Sydney, Australia.

system of equations and the initial and/or boundary conditions (BC). Following the learning process, the PDE coefficients and BC are assumed to be embedded into the neural network weights.

Another family of approaches involves solvers in which DL is used to approximate the solution of the PDE in an end-to-end fashion given the problem properties (coefficients, initial conditions or sources). That is, the network needs to generalize over some of the PDE properties, which is the more common scenario in computer vision applications. Some works employ Convolutional Neural Networks (CNNs) which are typically used for applications involving data on regular grids, like images and videos, or, discretized PDEs on a structured grid. Khoo and Ying [11] introduced a CNN named 'SwitchNet', for approximating forward and inverse maps arising from the time-harmonic wave equation, assuming plane-wave sources and 2-4 Gaussian scatterers in a constant medium. However, no neural network has been presented in the literature that effectively solves the Helmholtz equation at high wavenumber for rather general heterogeneous models and right-hand-sides.

In this work we utilize CNNs as preconditioners for the discretized Helmholtz equation. We wish to define a solver that generalizes over right-hand-sides and slowness models. We propose to use a CNN together with a classical multigrid approach—the Shifted Laplacian—which is efficient at removing only part of the error. We build on the similarity between a V-cycle and a U-Net [22] which is a multiscale CNN used for image-to-image mappings such as semantic segmentation, image denoising etc. Indeed, a geometric V-cycle can be naturally implemented using DL frameworks, suggesting that a U-Net architecture, like a V-cycle, can be used as a preconditioner for solving PDEs on a regular grid. To precondition the Helmholtz equation, we augment the U-Net with a classical approach, to complement the network and stabilize the solution process. We examine two preconditioners: one is a U-Net that acts as a non-linear coarse grid correction and is applied with a Jacobi iteration for pre-and post-smoothing. In the second formulation we apply alternating U-Net and SL multigrid V-cycle, which provides a more powerful smoothing and somewhat resembles the deflation preconditioner of [24, 5]. Both preconditioners are applied within a Krylov subspace method—flexible GMRES. We define our U-Net to generalize over a random right-hand-side and a random piece-wise smooth slowness model. We also suggest two upgrades to the scheme above: (1) an encoder-solver framework and (2) a mini-retrain approach. Both upgrades are described in details later, and can improve the standard U-Net tremendously.

## 2 Preliminaries and background

The heterogeneous Helmholtz problem is give by

$$-\Delta u(\vec{x}, \omega) - \omega^2 \kappa(\vec{x})^2 (1 - \gamma i) u(\vec{x}, \omega) = g(\vec{x}, \omega), \quad \vec{x} \in \Omega \tag{2.1}$$

The unknown $u(\vec{x}, \omega)$ represents the pressure wave function in the frequency domain, $\omega = 2\pi f$ denotes the angular frequency, $\Delta$ is the Laplacian operator and $\kappa(\vec{x})$ is the heterogeneous wave slowness - the inverse of its velocity. The source term is the right-hand-side of the system, $g$. The parameter $\gamma$ indicates the fraction of attenuation (we focus on the hardest case $\gamma = 0$), and $i = \sqrt{-1}$. As boundary conditions, we consider an absorbing layer [6], but PML [26] or [17] can be viable options as well. We consider the second order finite difference discretization of (2.1) on a uniform mesh of width $h$ in both. This leads to a system of linear equations

$$A^h \mathbf{u}^h = \mathbf{g}^h. \tag{2.2}$$

When discretizing the Helmholtz equation, we must obey the rule of thumb that at least 10 grid nodes per wavelength are used. This typically requires a very fine mesh for high wavenumbers, significantly increasing the number of unknowns and making the system (2.2) ill-conditioned, in addition to being indefinite. Hence, the solution of the problem can be challenging and require creative iterative solvers.

### 2.1 Shifted Laplacian multigrid

Eq. (2.2) is commonly solved by a preconditioned Krylov subspace method. The SL operator

$$Mu = -\Delta u - \omega^2 \kappa(\vec{x})^2 (\alpha - \beta i) u, \quad \alpha, \beta \in \mathbb{R} \tag{2.3}$$

is used to accelerate the convergence of a Krylov subspace method for solving (2.2). The preconditioning matrix $M^h$ is obtained from the discretization of (2.3), similarly to (2.1) yielding $A^h$. In this paper we focus on the pair $\alpha = 1$ and $\beta = 0.5$, which in [6] is shown to lead to a good compromise

between approximating (2.2) and our ability to solve the (2.3) using multigrid tools. The SL multigrid method is consistent and robust for non-uniform models, but is slow and for large wavenumbers. For details on the geometric multigrid method used here to approximately invert (2.3) please see Sec. A in our supplementary material.

# 3 Multigrid-augmented deep learning preconditioners

We utilize a U-Net architecture to accelerate the convergence of the Helmholtz solution. In order for the computational complexity to be cost-effective we define the architecture to be relatively small in DL standards (see Fig. 1), and yet demonstrate a significant acceleration to the solution. Our DL-based preconditioner is applied within the flexible GMRES Krylov method [23]. The flexible variant is important as our preconditioner is non-linear, hence also non-stationary.

The U-Net architecture [22] is a CNN for image-to-image tasks such as semantic segmentation. In our case, given a slowness model and a right-hand-side sampled on a grid (viewed as images), we wish to predict a solution for (2.2) on that grid. On its way down the hierarchy, the architecture interchangeably applies down-sampling convolutions and ResNet convolution blocks. Then, going up the hierarchy it applies up-sampling convolutions. The down- and up-sampling are applied using strided convolutions, similarly to multigrid transfer operators.

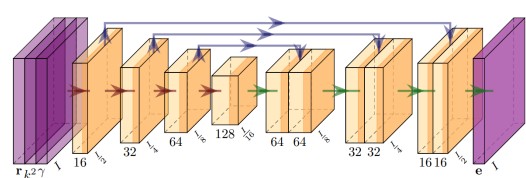

Figure 1: Our U-Net architecture.

**U-Net as a preconditioner accelerator** Since the medium can be very heterogeneous, the solution, even for a point source or a plane wave, can be highly complex at high wavenumber due to reflections and interference. Therefore, we do not pursue a solution via a single application of a network, but aim for a network to be applied several times in the form of a preconditioner. Specifically, we aim for the U-Net to complement the shifted Laplacian multigrid method, which is efficient at handling error modes with rather large absolute eigenvalues. We aim that the U-Net will focus on the these error directly, and it will be accompanied by classical smoothing via SL. This guides our training procedure, which is described in Sec. B in the supplementary material. Furthermore, the smooting is important because the Krylov method is sensitive to high residuals. Below we list two preconditioning schemes:

1. *U-Net as a coarse grid solver in a two-level cycle* In our first version, denoted as $M_{\text{JU}}$, we apply one pre- and post- Jacobi relaxations with a U-Net in between. Since the U-Net starts and ends with strided convolutions, this ends up being similar to a two-level cycle.

2. *U-Net as a deflation operator* A slightly more evolved option would be to apply U-Net, and afterwords to smooth the output using a SL V-cycle. More explicitly:

$$\text{Preconditioner } M_{\text{VU}}(\mathbf{r}) = \text{V-cycle}(\mathbf{e}_0 = \text{U-Net}(\mathbf{r}), \mathbf{r}). \tag{3.1}$$

This option somewhat resembles the concept of deflation operator studied in [25].

**Encoder-Solver: an encoder network as a preconditioner setup** The stand-alone solver described so far needs to generalizes on both the residual and the slowness. Obviously, the less we have to generalize on, the solution will be easier for the network. This is verified in our experiments—the network performs better when the slowness model is known during training (see Sec. C.2). Hence, we wish to exploit the following: when we solve an equation using several iterations, the slowness model remains the same during those iterations—the U-Net is repeatedly applied with the same model for different residuals. To exploit this, we propose to split the generalization over the residual and slowness model into two "encoder" and "solver" networks. The encoder network receives only the slowness model as input, and prepares feature vectors that include the slowness model information. Those vectors are computed only once at the beginning of the solution process, and are fed into the solver network at every iteration in the solution time. The solver network is applied at every iteration, with the readily available feature vectors inside it. This idea resembles a preconditioner setup that is applied once before the solution process, and is adequate for any right hand side.

**Mini re-training** Our last approach aims to further improve the solver U-Net by a mini retraining procedure at solve time for the given slowness model. That is, given a slowness model $\kappa^2$ we apply a

few training iterations over a randomly generated data set of residuals to improve the U-Net weights. This is similar to the process obtained in PINNs, and is motivated by our relatively better performance for known models compared to the case where we generalize over the models. This option would fit cases where we need to solve multiple linear systems with the same slowness model but for many right-hand-sides, like when solving inverse problems, where the medium needs to be estimated.

## 4    Numerical Results

In this section we evaluate the performance of the preconditioner together with flexible GMRES for solving the Helmholtz equation for random right-hand-sides. In particular, we use a restarted and flexible Block-GMRES(10), with a subspace of size 10, solving 10 different right-hand-sides together. We consider it to be the preconditioner test. Our code is written in the Julia language [3], utilizing the Flux DL framework [10]. Unless stated otherwise, the test cases that we present are for (2.1) defined on a 2D domain of $[0, 1]^2$, discretized on a nodal regular grid of $128 \times 128$ cells, and a wave frequency defined by $\omega = 20\pi$, which leads to about 12.5 grid points per wavelength. We use an absorbing boundary layer of width 10. We compare three preconditioners: The SL V-cycle $M_{\mathsf{V}}$, $M_{\mathsf{JU}}$ and $M_{\mathsf{VU}}$ for Eq. (3.1). We evaluate the methods as preconditioners by their average convergence factor denoted by $\rho$, calculated over the number of iterations required to reduce the residual norm by a factor of $10^6$. For the SL V-cycles we use a shift of $\beta = 0.5$, 3 levels, 1 pre- and post- damped Jacobi smoothing with damping of 0.8, and a coarsest grid solution of GMRES(10) as in [4].

We now present the ability of the proposed methods to generalize for different slowness models and right-hand-sides. In these experiments, the training phase was performed on a set of $20,000$ pairs of $(\mathbf{r}, \mathbf{e})$ produced as described in Sec. B.1, and $\kappa^2$ were randomly generated from a collection of the $50,000$ images in the CIFAR10 data-set. We train the network on a grid of $128 \times 128$ for $\kappa^2 \in [0.25, 1]$, but also examine the performance on larger grids and frequencies. In particular, we solve (2.2) for random right-hand-sides on grids of size $256 \times 256$ and $512 \times 512$ with $\omega = 40\pi, 80\pi$, respectively , using a network trained on residuals and slowness models of size of $128 \times 128$ only. In supplementary Sec. C.3 we also show generalization for $\kappa^2 \in [0.5, 1]$.

Figure 2 summarizes the results for all our three methods: stand-alone preconditioner, encoder-solver framework, and the mini-retrain approach. Overall, it is clear our CNN can generalize over the right-hand-sides (residuals), and over the slowness models. The stand-alone solver significantly out-perform the SL solver, with $M_{\mathsf{VU}}$ typically outperforming $M_{\mathsf{JU}}$. Our encoder-solver framework, where two CNNs are used, indeed upgrades the stand-alone U-Net in all cases, and particularly when we test at a larger problem size than the network is trained on. In addition, our mini-retraining phase consistently yields another upgrade to the performance. While our preconditioner is not necessarily superior than traditional ones in terms of FLOPs, it can be applied more efficiently on hardware. Furthermore, CNN models can be significantly compressed without harming performance.

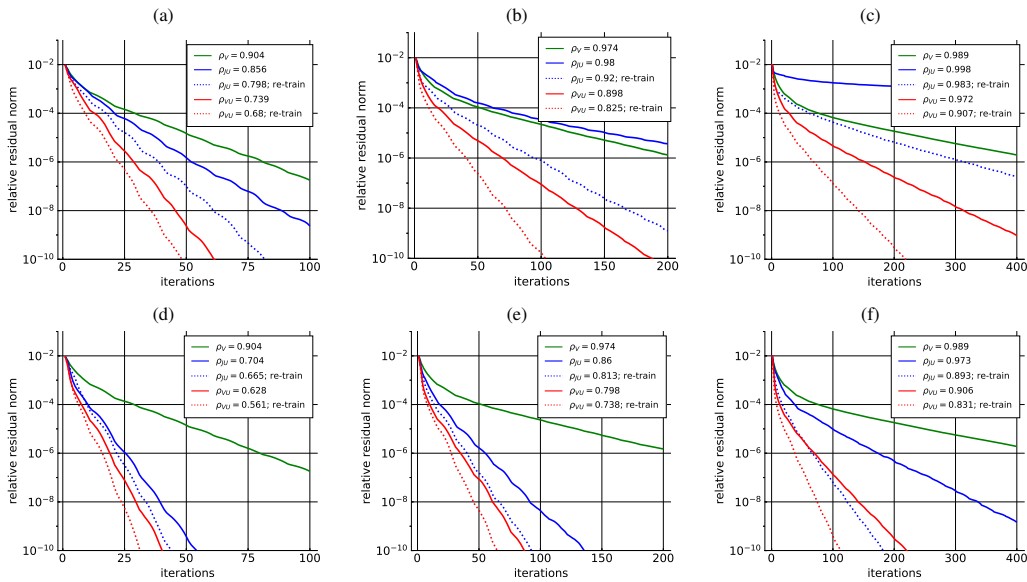

Figure 2: The convergence history of the solutions of 2.2 for random RHSs. First row: stand-alone U-Net for problem of size (a) $128 \times 128$, (b) $256 \times 256$ and (c) $512 \times 512$. Second row: the encoder-solver U-Nets the same experiments. Solid and dotted lines mark the preconditioners without and with the mini-retraining phase at solve time, respectively.

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

# Supplementary Material

## A  Background

### A.1  Geometric multigrid

In this section we describe the geometric multigrid approach that we use for solving (2.2). Generally, multigrid methods are used to iteratively solve linear systems like (2.2) defined on a fine grid $\Omega^h$ using a hierarchy of grids. In a nutshell, in a two level setting, we project the fine level problem onto a coarser grid, solve the problem on that coarser grid, and then use the coarse solution to correct the solution on the original fine grid. This is repeated recursively to form a V-cycle, which is applied iteratively to solve the problem. Geometric multigrid can be implemented using CNN constructs quite naturally, resulting in a structure that resembles the well-known U-Nets.

More explicitly, multigrid methods are based on two complementary processes: relaxation and coarse grid correction. The relaxation is obtained by a standard method like Jacobi or Gauss-Seidel, which is only effective at reducing part of the error[2]. The remaining error, called "algebraically smooth", is typically spanned by the eigenvectors of $A^h$ corresponding to small eigenvalues (in magnitude), i.e., vectors $\mathbf{e}^h$ s.t.

$$\left\| A^h \mathbf{e}^h \right\| \ll \left\| A^h \right\| \left\| \mathbf{e}^h \right\|. \tag{A.1}$$

To reduce such errors, multigrid methods use a coarse grid correction, where the error $\mathbf{e}^h$ for some iterate $\mathbf{u}^h$ is estimated by solving a coarser system and interpolating its solution:

$$A^H \mathbf{e}^H = \mathbf{r}^H = I_h^H(\mathbf{g}^h - A^h \mathbf{u}^h), \quad \mathbf{e}^h = I_H^h \mathbf{e}^H.$$

Here, the matrix $A^H$ is an approximation of $A^h$ on a coarser mesh $\Omega^H$, obtained for width $H = 2h$. To interpolate the solution from coarse to fine grids we choose the bi-linear interpolation operator $I_H^h$ which is suitable for the problem because the Laplacian operator in (2.3) is homogenous. The restriction operator $I_h^H$, dubbed "full-weighting", projects the fine-level residual to the coarse grid. Since the coarse problem is still too large to solve, the process of relaxation and coarse grid correction is applied recursively resulting in the V-cycle algorithm given in Alg. 1.

---

**Algorithm 1:** Multilevel V-cycle: $\mathbf{v}^h \leftarrow$ V-cycle($\mathbf{v}^h, \mathbf{f}^h$)

---

Relax $v_1$ times on $A^h \mathbf{v}^h = \mathbf{f}^h$ with $\mathbf{v}^h$ as an initial guess ;
**if** *deepest level* **then**
 |  $\mathbf{v}^h \leftarrow$ Solve the system $A^h \mathbf{v}^h = \mathbf{f}^h$ directly;
**else**
 |  $\mathbf{f}^{2h} \leftarrow I_h^{2h}(\mathbf{f}^h - A^h \mathbf{v}^h)$;
 |  $\mathbf{v}^{2h} \leftarrow$ V-cycle($\mathbf{v}^{2h} \leftarrow \mathbf{0}, \mathbf{f}^{2h}$);
 |  $\mathbf{v}^h \leftarrow \mathbf{v}^h + I_{2h}^h \mathbf{v}^{2h}$;
**end**
Relax $v_2$ times on $A^h \mathbf{v}^h = \mathbf{f}^h$ with $\mathbf{v}^h$ as an initial guess;

---

In Alg. 1 one has to choose the number of levels in the V-cycle. Unlike other scenarios, the smooth error modes of the Helmholtz operator are oscillatory at high wavenumber, and cannot be represented well on very coarse grids. Hence, the performance of the solver deteriorates as we use more levels. For example, the results in [4] show that the best performance is achieved using three levels only, and the authors suggest to use GMRES as a coarsest grid solver.

### A.2  Geometric multigrid using CNN modules

In our work, both the CNN and the multigrid preconditioner are used interchangeably in the same framework, using a GPU back-end. We implement the multigrid method using CNN components, in order to enable a good integration of the V-cycle with the CNN in a unified framework within the

---

[2]In the case of the Helmholtz system, such relaxation methods do not converge due to the indefiniteness of $A^h$, yet they are effective at smoothing the error using 1-2 iterations only.

preconditioner component. The idea is based on the close connection between U-Nets and multigrid V-cycles.

The Helmholtz operator needed for the residual calculation and the Jacobi relaxation is executed using a convolution operator according to the discretized Laplacian operator

$$\Delta_h = -\frac{1}{h^2} \begin{bmatrix} 0 & -1 & 0 \\ -1 & 4 & -1 \\ 0 & -1 & 0 \end{bmatrix}$$

using a fixed-weights $3 \times 3$ kernel plus an element-wise multiplication with the mass term in (2.3) or (2.1), i.e., whether with an artificial shift or not. The high order discretization operators in [28] can also be obtained similarly.

The geometric multigrid transfer operators, the restriction (coarsening of the mesh) and prolongation (interpolation), are realized by convolution operators with the fixed kernels

$$K_{I_h^H} = \frac{1}{16} \begin{bmatrix} 1 & 2 & 1 \\ 2 & 4 & 2 \\ 1 & 2 & 1 \end{bmatrix}, \quad K_{I_H^h} = \frac{1}{4} \begin{bmatrix} 1 & 2 & 1 \\ 2 & 4 & 2 \\ 1 & 2 & 1 \end{bmatrix}, \tag{A.2}$$

that correspond to the full-weighting and bi-linear operators, respectively. Both kernels are applied with a stride of 2. For the restriction, this results in a coarse mesh $\Omega^H$. For the prolongation, we use a transposed strided convolution with $K_{I_H^h}$, so that the mesh is refined. The solution for the coarsest mesh problem is done using GMRES with diagonal Jacobi preconditioner [4]. In addition, all these convolutions are applied with zero padding of size 1, which is consistent with the Dirichlet boundary condition (the ABC is obtained through the mass term). Neumann BC can be obtained by using reflection or replication padding in CNN frameworks, which result in first and second order Neumann BC, respectively.

## B    Method: training, data generation and augmentation

### B.1    Training and data augmentation

In order for the U-Net to be efficient at solve time as a preconditioner, it must practice or learn on GMRES residual vectors that are as close to the solve time GMRES residuals as possible. Let

$$\mathbf{e}^{net} = \text{U-Net}(\mathbf{r}, \kappa^2; \theta) \tag{B.1}$$

be the forward application of the network for a given right-hand-size $\mathbf{r}$ and slowness model $\kappa(\mathbf{x})^2$. As before, $\theta$ denote the collection of network weights (convolutions, biases, etc.) that we wish to learn. Since we wish to complement smoothing processes, in our supervised learning setup, we aim to minimize the error with respect to multiple residuals and models. That is, given training data triplets $\{(\mathbf{e}_i^{true}, \mathbf{r}_i, \kappa_i^2)\}_{i=1}^m$, where $A^h \mathbf{e}_i^{true} = \mathbf{r}_i$, we minimize

$$\min_\theta \frac{1}{m} \sum_{i=1}^m \|\text{U-Net}(\mathbf{r}_i, \kappa_i^2; \theta) - \mathbf{e}_i^{true}\|_2^2, \tag{B.2}$$

which is the mean $\ell_2$ error norm, also known as the mean squared error (MSE). The training phase is performed by a stochastic gradient descent optimizer, where in each epoch we sweep through all the training examples in batches and varying order. Specifically, we use the ADAM optimizer [12] with a variable step size.

For supervised learning, we have to provide the network with the corresponding error solution for each residual vector and slowness model. To have such triplets, in some cases one needs to invest a lot of time and resources in preparing the training-set, i.e., solve the PDE many times. However, since we have a linear system here, we can simply generate a random $\mathbf{e}_i^{true}$, and easily compute the residual. But, we wish to create a data-set that is as close as possible to the situation in the real solution using GMRES, and if we simply take the GMRES residual vectors, we will not have a corresponding error without solving the system. To obtain that we need to generate data with rather smooth vectors in varying "smoothness levels", since GMRES and V-cycles are smoothing operations. Hence, we propose the following procedure for which, in our experience, the training error predicts the final

solution's convergence factor quite accurately. First, we generate a random normally distributed $\mathbf{x}_i$ and compute $\mathbf{b}_i = A^h \mathbf{x}_i$. Then, we apply a random number of GMRES iterations with SL V-cycle as preconditioner, starting from a zero vector, to get $\tilde{\mathbf{x}}_i$:

$$\tilde{\mathbf{x}}_i = \text{GMRES}(A^h, M = \text{V-cycle}, \mathbf{b_i}, \mathbf{x}^{(0)} = \mathbf{0}, iter \in \{1, ..., 10\}). \tag{B.3}$$

Following that, we compute a residual $\mathbf{r}_i = \mathbf{b}_i - A^h \tilde{\mathbf{x}}_i$ and let $\mathbf{r}_i$ be the network input. The true error that we wish the network to recover is given by

$$\mathbf{e}_i^{true} = \mathbf{x}_i - \tilde{\mathbf{x}}_i. \tag{B.4}$$

Beyond the procedure above, we insert $\hat{r}_i = h^2 \mathbf{r}_i$ as input to the network. This causes the network's input and output to be of the same order of magnitude[3], and because the system is linear, this change has no effect on the accuracy. Furthermore, to avoid over-fitting and achieve better generalization, we augmented the training examples using a linear combination of the errors and residuals to expand the set. This summarizes the training data generation, given slowness models. The training set and the validation set are manufactured in the same way, and we saw no difference in the loss between these two.

Overall, our network has 4 input channels of the size of the grid. These are the real and imaginary parts of the residuals $\mathbf{r}_i$, the real (squared) slowness model $\kappa_i(\mathbf{x})^2$, and the real attenuation model $\gamma(\mathbf{x})$. Although we do not generalize of $\gamma(\mathbf{x})$, we found it slightly beneficial to include it as input, as it allows the network to have a different behaviour at the grid's boundaries.

**Training details** The training process for all the U-Nets reported in this paper is performed by the ADAM optimizer [12], and includes 80-120 epochs. The initial learning rate is $10^{-4}$ and the initial mini-batch size is 20. Along the epochs we reduce the learning rate and increase the mini-batch size. The schedule of these changes as well as the amount of epochs we apply vary a bit between the different experiments and are adjusted by cross validation on the test set. In the majority of the cases, after $t_0 = 48$ epochs we divide the initial learning rate by 10 and double the mini-batch size. The next changes of the learning rate and mini-batch size occur after every $t_i = \frac{t_0}{2^i}$ epochs. That is, the changes occur at epocs 48, 72, 84, 90 etc, until the learning rate essentially reaches zero. These settings can be used as default parameters for all the experiments that we show.

**Remark 1** *As an alternative to the error loss in* (B.2)*, one may use the residual norm* $\|A^h \mathbf{e}_i^{net} - \mathbf{r}_i\|_2$*, so that the training can be unsupervised [1]. This is a viable option, but it can be a bit tricky, since high errors can in fact have low residuals—these are "algebraically smooth errors". Since we can eliminate those errors with V-cycles or Jacobi iterations, we wish to let the network focus on those smooth modes and not on the whole spectrum. We note that empirically, our experience in augmenting the error norms in* (B.2) *with residual norms generally yields inferior performance than focusing on the errors only during training.*

## B.2 Slowness models

Solving the Helmholtz equation for a uniform slowness model is not difficult, as it can be performed using a Fourier transform. When trying to solve this equation over a non-uniform media, the difficulty increases significantly, especially if there are sharp jumps or large ratios between the smallest and highest slowness. Here, we want the network to succeed in different types of models, smooth and non-smooth, that faithfully represent reality. In certain applications (e.g., geophysical or medical imaging) one may have slowness models of a certain typical distribution coming from the application domain. E.g., in geophysical imaging, one typically has a layered model with higher velocity values deeper in the earth. In machine learning tasks, if we wish for an algorithm to generalize on an input, it is best if we let the algorithm learn on inputs from the same distribution as in the real life application. That is a large part of the advantage of data-driven approaches. Usually, however, to obtain a good generalization we need a large data set of inputs—a collection of slowness models in our case.

Here, we use a rather general slowness model collection to demonstrate our method. The models that we use for the U-Net training, are created from a large set of natural images CIFAR-10 [13]. This data set is usually used as image classification benchmark and contain 50K images of resolution $32 \times 32$

---

[3]The magnitude of the input and output are important because of how certain default parameters are chosen in DL frameworks—e.g., initialization of weights and biases and scales of activation functions. For this reason, the input is usually normalized, and batch normalization is applied throughout the network.

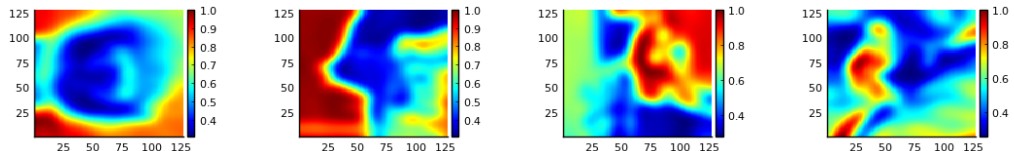

Figure 3: Four examples for the $\kappa(\mathbf{x})^2$ models generated from the CIFAR10 image collection. Each model is a random image from the set, enlarged to a grid of 128 by 128 pixels, smoothed using a Gaussian kernel and normalized to the range of [0.25,1].

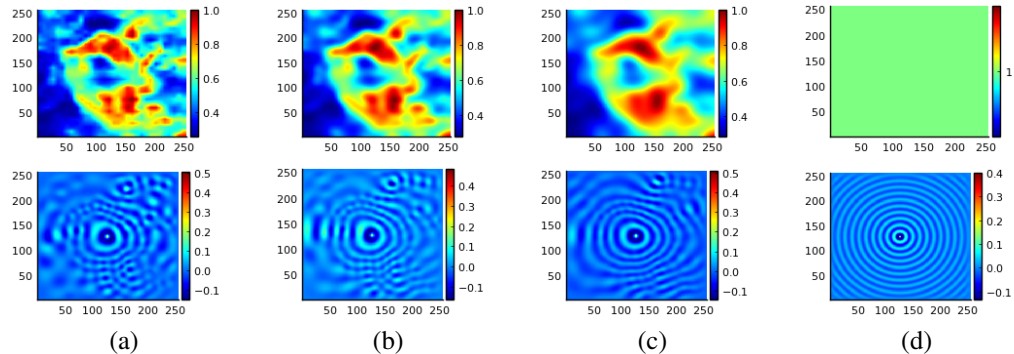

Figure 4: Illustration of smooth levels of $\kappa(\mathbf{x})^2$, and their effect on the full solution of the Helmholtz equation for a point source. In the upper row we show the model $\kappa^2$ for each test. The leftmost, (a), is an image from the CIFAR10 data-set, enlarged to a $256 \times 256$ grid, and the two inner figures, (b) and (c), are the same model after a Gaussian smoothing with kernel sizes of 5 and 10, respectively. The rightmost, (d), is the constant homogeneous model. On the bottom row we show the full solution for the corresponding model in the first row.

pixels. We use this data set and manipulate the images to become slowness models in order to have a significant amount of training models to learn from. In particular, we use a training set of $20,000$ random sample images. To transform the natural images into slowness models, we first enlarge them to the grid size using a bi-linear interpolation, then smooth them with a single application of a Gaussian smoothing kernel, and normalize the values to the range of $[0.25, 1]$. Fig. 3 shows four different models produced from the CIFAR-10 image collection. Fig. 4 presents how the smoothness level of the model, $\kappa^2$, affects the difficulty of the solution. It is evident that as the medium is less smooth, the solution is more complex and is harder to model by a neural network. Similarly, the range of slowness values in the model also affects the the complexity of the solution. Fig. 5 illustrates how the solution is again more complex as the contrast in the slowness model is larger.

## C    Additional Numerical Results

In addition, we present the relative error norm plot throughout the training period, which shows the value of the loss function (B.2) vs. the epoch number. Ideally, the asymptotic value of the relative

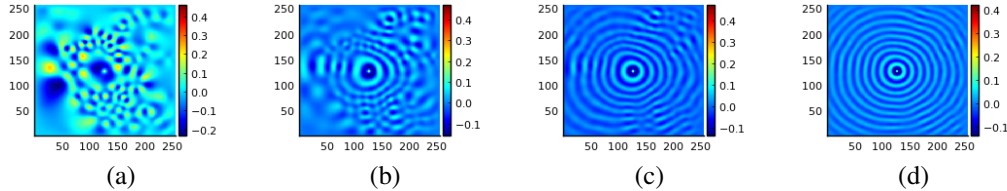

Figure 5: Illustration of the effect of the value ranges of $\kappa^2$ on the full solution of the Helmholtz equation for the point source problem. The leftmost, (a) is a full solution for kappa normalization between 0 and 1, (b) $\kappa^2 \in [0.25, 1]$, (c) $\kappa^2 \in [0.5, 1]$ and (d) $\kappa^2 \in [0.75, 1]$.

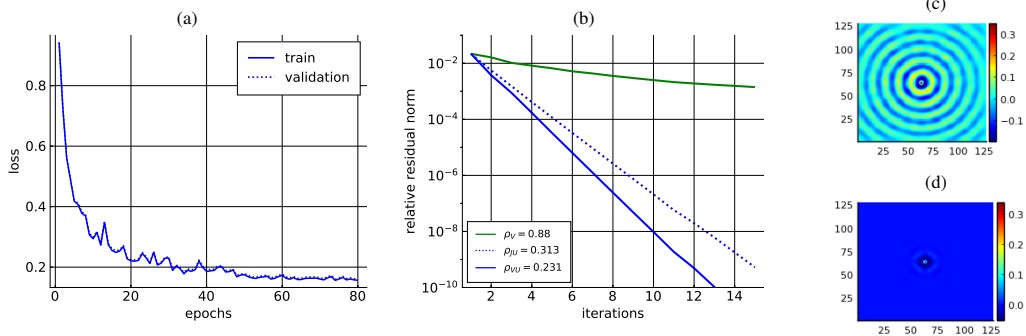

Figure 6: Experiments for a homogeneous slowness model. (a) the training plot. (b) the residual drop history of the GMRES method. $\rho_V$,$\rho_{JU}$,$\rho_{VU}$ are the convergence factors of the preconditioners $M_V$, $M_{JU}$ and $M_{VU}$, respectively. (c) shows the real part of the result of a single U-Net operation on a point source, compared to the result of a single V-cycle for the same input in (d).

training error should indicate the real convergence factor at solve time using GMRES. It is indeed more-or-less so thanks to the varying smoothing levels that we apply for our training residuals in Eq. (B.3). Using this plot, we can also estimate over-fitting and adjust the hyper-parameters (learning rate and mini-batch size, for examples).

### C.1 Performance of a homogeneous model

In the homogeneous test case, we solve the system (2.2) for a uniform model, known at training. That is, the training is performed on random residual and error vectors only as described earlier. Fig. 6 describes the results of the proposed method for this case. The average relative error, i.e. the minimum estimate value of the loss function (B.2), of the training and test sets at the end of the training process is 0.18. The GMRES method with U-Net-based preconditioner as described in Eq. (3.1) converges within 10 iterations with convergence factor $\rho_{VU} = 0.231$ compared to 0.88 using only V-cycles. Our cheaper $M_{JU}$ preconditioner, which uses only our U-Net and Jacobi steps converges within 12 iterations with convergence factor $\rho_{JU} = 0.313$. These are very favorable convergence factors given how poorly V-cycles behave for a homogeneous model. On the other hand, homogeneous problems can be efficiently solved using fast Fourier transform.

### C.2 Performance on a single heterogeneous model

In this section we examine our method for a single heterogeneous slowness model, known during the training phase. In this scenario, the network can learn weights specific to that model. As shown in Fig. 7, for a model normalization of $\kappa^2 \in [0.5, 1]$ we obtained convergence factors of $\rho_{VU} = 0.27$ and $\rho_{JU} = 0.383$, which are close to the performance on the uniform model. For the more difficult case of $\kappa^2 \in [0.25, 1]$, we obtain $\rho_{VU} = 0.41$, $\rho_{JU} = 0.456$. Furthermore, in plot (c) of Fig. 6 and in the bottom plots (c) and (e) of Fig. 7, we show the results of a single iteration of Eq. (3.1) on a centralized point source, a problem that is not included in the training set. It is evident that U-Net has indeed learned to produce a global solution, and the integration with the V-cycle is intended to "clean" or smooth the solution and ensure convergence.

### C.3 Performance on generalized slowness models

As we showed in the previous section, our U-Net can learn to solve (2.2) quite efficiently for a given heterogeneous slowness model. However, the training of the U-Net is quite expensive to apply at solve time. In a more applicable approach for the realistic world, we wish the neural network to work and generalize on models it had not seen in the training phase. Therefore, we also present the ability of the proposed method to generalize for different models. Table 1 presents the results for the encoder-solver setting. Regardless of the specific preconditioner, we see that if the networks are trained on a single model, they exhibit the best performance. However, they fail to generalize on slightly different cases—even for the easier normalization of the same model (row (a) vs row (b) in

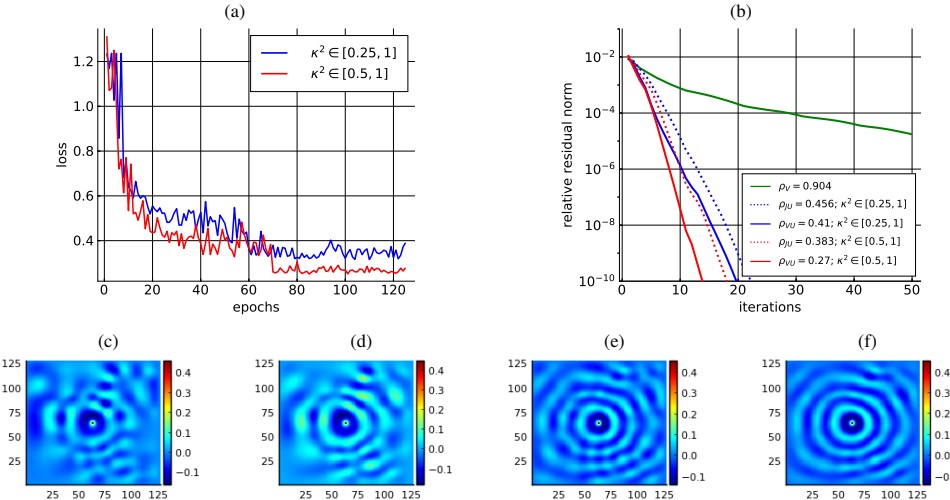

Figure 7: Experiments for a known heterogeneous slowness model. We compare models for which $\kappa^2 \in [0.5, 1]$, and $\kappa^2 \in [0.25, 1]$. (a) shows the convergence in the training phase and (b) shows the results of the three preconditioner tests. The bottom figures describe the real part of the results of a single U-Net operation, (c) for $\kappa^2 \in [0.25, 1]$ compared to the real part of the full solution in (d), and (e) for the case of $\kappa^2 \in [0.5, 1]$ compared to the full solution (f).

|  |  | $M_V$ | Single model | | General model | | Re-trained | |
|  |  |  | $M_{VU}$ | $M_{JU}$ | $M_{VU}$ | $M_{JU}$ | $M_{VU}$ | $M_{JU}$ |
|---|---|---|---|---|---|---|---|---|
| (a) $\kappa^2 \in [0.25, 1]; n = 128$ | $\rho$ | 0.904 | 0.41 | 0.456 | 0.628 | 0.704 | 0.561 | 0.665 |
|  | $T$ | 137 | 16 | 19 | 30 | 40 | 24 | 34 |
| Generalization for cases not seen in training | | | | | | | | |
| (b) $\kappa^2 \in [0.5, 1]; n = 128$ | $\rho$ | 0.867 | 0.827 | fail | 0.516 | 0.573 | 0.432 | 0.498 |
|  | $T$ | 97 | 73 |  | 21 | 25 | 17 | 20 |
| (c) $\kappa^2 \in [0.25, 1]; n = 256$ | $\rho$ | 0.974 | 0.958 | fail | 0.798 | 0.86 | 0.743 | 0.811 |
|  | $T$ | 391 | 240 |  | 62 | 92 | 46 | 67 |
| (d) $\kappa^2 \in [0.5, 1]; n = 256$ | $\rho$ | 0.946 | 0.887 | fail | 0.681 | 0.727 | 0.583 | 0.665 |
|  | $T$ | 208 | 116 |  | 36 | 44 | 26 | 34 |

Table 1: Performance comparison of the U-Net-based preconditioners. $\rho$ is the convergence factor and $T$ is the number of iterations required to reduce the residual norm by a factor of $10^6$. (a) summarizes the numerical results of the method on new examples equal in difficulty to the training set. Rows (b),(c) and (d) describe the performance of the same U-Nets on problems of a different size or with a different level of contrast in the slowness model. The "single model" column demonstrate the performance when the U-Net is trained using a single heterogeneous model. In the "general model" column the network is trained on multiple slowness models. The "re-trained" column describes the performance after mini-retraining.

the single model column). If we train the networks to generalize over slowness models, it can also generalize on the problem size and slowness contrast. All cases are improved using mini-retraining.

### C.3.1 The influence of the Krylov subspace and block sizes

In this set of experiments we examine the influence of the subspace sizes that the GMRES method uses. We examine the influence of both the subspace size and the number of RHSs solved together (block size). The experiment corresponds to the $256 \times 256$ experiment shown in Fig 2 (e). Fig. 8 summarizes the results. It is obvious that whether by multiple RHSs or by subspace size, increasing the effective Krylov subspace can improve the performance of the solver significantly. That is in contrast to the SL V-cycle that is not improved by the same magnitude. This shows that there is more potential to the U-Net based preconditioners that shown earlier, since in addition, we typically did not

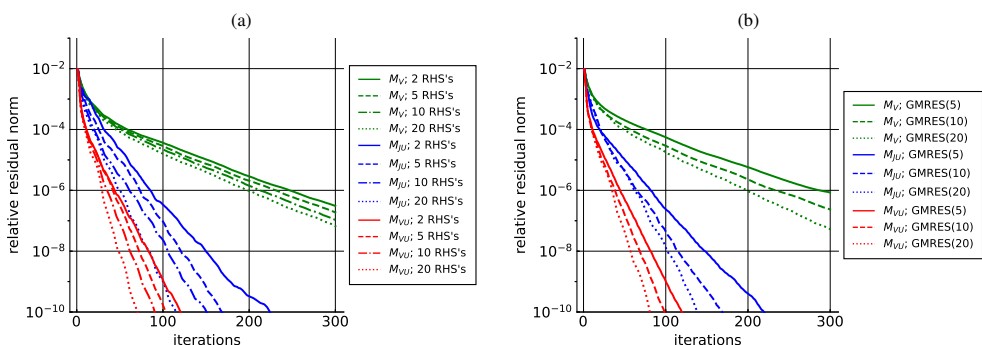

Figure 8: The influence of the Krylov subspace and block sizes. On the left we see the influence of the block size (convergence for a number of right-hand-sides solved together by block flexible GMRES(10)). On the right we see the influence of the subspace size on the convergence factor.

reach the training error drop in the preconditioner test, indicating that GMRES stabilizes the solution on the one hand, but also slows the preconditioners due to high residuals.

