# OpenReview forum: "Multigrid-augmented deep learning preconditioners for the Helmholtz equation"
_NeurIPS.cc/2021/Workshop/DLDE — DLDE Workshop -- NeurIPS 2021 Poster_

### Official Review · Reviewer_H8oS · 2021-10-03
**Interesting  approach but missing reference**

**Confidence:** 5

**Review:**

Helmholtz equation at high wave numbers is hard. This paper combine multigrid ingredients with convolutional neural networks (CNNs) to form a preconditioner which is applied within a Krylov solver and gives impressive convergence result.
1. The paper only shows the convergence figure with respect to the iterations
2. Missing Reference:
The review finds out the paper missed several approaches using NN as a preconditioner, the author should cite them and discuss the relationships, the missed citations includes
[1] Tompson J, Schlachter K, Sprechmann P, et al. Accelerating eulerian fluid simulation with convolutional networks[C]//International Conference on Machine Learning. PMLR, 2017: 3424-3433.
[2] Hsieh J T, Zhao S, Eismann S, et al. Learning neural PDE solvers with convergence guarantees[J]. arXiv preprint arXiv:1906.01200, 2019.
[3] ....
3. The reviewer is confused by the description of the " Mini re-training " in the paper, It is said it's common in PINN, but I can't see a reference?
4. It's also interesting to do the similar experiment in this draft https://openreview.net/pdf?id=mueJ9ZjrfOF

Overall, the paper proposed a fast solver for Helmholtz equation at high wave numbers and gives a excellent technical report.  However lack of the discussion with the previous work to distinguish the difference between different way of using NN as a preconditioner makes the novelty of the paper not significant.

**Score:**

3: Good paper

---

### Official Review · Reviewer_Edmf · 2021-10-11
**A valid contribution in the use of the U-net preconditioners to solution of the Helmoltz equation**

**Confidence:** 1

**Review:**

The paper proposes a method for solving the Helmoltz equation by formulating a hybrid multi-grid / CNN solution by using a CNN-based approach as a pre-conditioner in the multigrid approach.  The U-Net architecture that was developed in computer vision resembles a V-cycle and it is proposed to us that in two ways as a pre-conditioner.  First, Unet is applied with a Jacobi iteration for smoothing; second U-net is applied in a a shifted Laplacian multigrid v-cycle.  Numerical results are shown; the new methods out-perform the traditional SL-solver in terms of error/convergence.

The contribution of the U-net preconditioners to solution of the Helmoltz equation seems to be a valid contribution.  The method is evaluated to show improvement over a more traditional optimizer.  The presentation of the paper seems to be poor though, its not self-contained in the 4 pages.  The actual way that the Unet is integrated into multigrid and trained is not even given; its in the appendix.  I think its important enough for the main paper.  The plots (its unclear how rho is computed) and details of the numerical results could be simplified; where is the standard SL solver in these plots?  Seems only the two versions of Unet are shown.  I believe the method contribution is good for this workshop, however, the presentation could be improved; it wasn’t a pleasant read.

**Score:**

2: Borderline paper

---

### Official Review · Reviewer_a5ZY · 2021-10-12
**A nice application of deep learning for data-driven solution of a challenging class of parameterized PDEs**

**Confidence:** 4

**Review:**

Summary

The authors study the Helmholtz equation in the case of high wavenumber and spatially varying coefficients. This is supposedly a challenging problem to solve classically (i.e., using existing state-of-the-art non-DL methods); although I have no prior experience with this specific PDE problem, the features that make it challenging (multiscale behaviour, complex-valued functions, indefinite operators/matrices, general ill-conditioning) are well-explained. Moreover, relevant citations to current SOTA methods appear to be included.

Motivated by recent success in using deep learning to study PDEs, the authors propose to use CNNs to produce better preconditioners for a multigrid solution method involving a Krylov solver. A few related variants of this method are presented. The authors use the U-Net architecture as a starting point for their methods, and analogies are drawn between this architecture and the behaviour of classical multigrid methods.

Discussion:

In general, the background discussion would have benefited from some additional details (especially to make it more accessible to non-experts in the specific problems/methods being discussed). It should be stated earlier than the ‘wave slowness models’ refer to the kappa function, and the ‘right-hand-side’ should be called the g function and maybe referred to as a source term (if I indeed understand the meaning of these correctly).

Line 23: It is stated that the SL preconditioner (the SOTA) “is not scalable”. With respect to what problem parameters does the SL preconditioner scale poorly? Can the authors measure the scaling of their proposed methods with respect to those same parameters?

Lines 36-45: This discussion would likely benefit from additional discussion of similar works in the literature. The authors may wish to consider the following works and citations within/thereof.
This other work by Khoo et al. is very related to the authors efforts to deal with heterogenous PDE parameters: https://arxiv.org/pdf/1707.03351.pdf.
This work by Geist et al. and related works provide insights into the performance of this class of methods. https://link.springer.com/article/10.1007/s10915-021-01532-w

The authors may also wish to consider work that sits between the two classes of DL-based PDE solvers they identified. That is, many authors have considered deep learning methods that solve PDEs directly (i.e., without training on precomputed solutions), but also solve these problems as functions of problem parameters. The DGM paper was one of the first to do this, I believe: https://arxiv.org/abs/1708.07469.

Line 166: What is meant by ‘it can be applied more efficiently on hardware’? Do the authors mean to say that DNNs can more readily leverage GPU acceleration that traditional methods like SL preconditioners?

Line 167: I would caution the authors against assuming without evidence that their networks can be compressed without harming performance. Geist et al. (cited above) and many others have discussed the fundamental approximation theoretic limits of representing PDE solutions with neural networks encoded in finite precision numbers. Although I would say that the conversation around accuracy vs. memory consumption in DLDE is far from over, I think it is reasonable to say that regressing complicated continuous-valued functions (like PDE solutions) is fundamentally different than solving classification tasks. In particular, the boundaries of classification tasks are not necessarily uniquely defined on a given training set, and this is likely a factor in the very impressive compression results attained on those deep learning applications (which I presume the authors are alluding to with this comment). Nonetheless, the question of network compression is a very interesting one from both theoretical and applied perspectives, and the potential advantages of low-memory PDE solutions is worthy of exploration in my opinion.

Throughout the paper, the authors discuss the importance that their DL solvers generalized (across residuals, slowness, right-hand-sides, etc.). Although the success of their method implies that some generalization is indeed occurring, a more careful discussion of this topic might be a nice extension of the work. For instance, if the networks are trained only on kappa functions with many small-scale features (like Fig 4a), how do they perform on test problems with lower-frequency kappa functions (like Fig 4d)?

Regarding mini re-training and Remark 1: The result that PINN-style training led to better performance in Fig2 but worse performance in Remark 1 is likely of interest to the DLDE/SciML community more broadly. It might be worth explaining these results in more detail.

The idea to convert CIFAR10 images to wave slowness models is quite interesting. In contrast, others have often used various parameterized function spaces (e.g., see the various spaces studied by Geist et al. 2021). I would be curious to see more discussion of the nature of the function space generated by processing CIFAR10 in this way; how hard are the PDEs generated in this way?

As a minor note, the submission would have benefitted from additional proofreading. There are a significant number of typos, missing/extra words, etc., and in places the language is a bit too imprecise/informal. Some of this is to be expected in extended abstracts of restricted length, but nonetheless this could have been handled a bit better.

Overall, this paper is an interesting application of DL to solve DEs, and I recommend it be accepted to the workshop. As written, much of the submission is not readily accessible to readers who aren’t already specialized in solving heterogeneous Helmholtz with multigrid methods (or something similar). I recommend that, for their poster, the authors make an effort to highlight the aspects of their work that will be of particular interest to the DLDE community. Insofar as details specific to multigrid methods and/or the Helmholtz equation are important to the discussion, these should be explained more clearly for non-experts in those topics. Similarly, a brief discussion of the physics of the heterogeneous/high-frequency Helmholtz equation (and in what applications it may arise) would be nice.


**Score:**

4: Very good paper

---

### Decision · Program_Chairs · 2021-10-16

**Decision:**

Accept (Poster)

**Comment:**

Reviewers seem to mostly agree on the acceptance of this article.